# Novel Anti-Aging Benzoquinone Derivatives from *Onosma bracteatum* Wall

**DOI:** 10.3390/molecules24071428

**Published:** 2019-04-11

**Authors:** Umer Farooq, Yanjun Pan, Dejene Disasa, Jianhua Qi

**Affiliations:** College of Pharmaceutical Sciences, Zhejiang University, Yu Hang Tang Road 866, Hangzhou 310058, China; 11519039@zju.edu.cn (U.F.); 21719029@zju.edu.cn (Y.P.); 11719053@zju.edu.cn (D.D.)

**Keywords:** anti-aging, *Onosma bracteatum*, structure elucidation, benzoquinone, replicative lifespan, K6001

## Abstract

The aim of this study was to investigate anti-aging molecules from *Onosma bracteatum* Wall, a traditional medicinal plant used in the Unani and Ayurvedic systems of medicine. During bioassay-guided isolation, two known benzoquinones, allomicrophyllone (**1**) and ehretiquinone (**2**) along with three novel benzoquinones designated as ehretiquinones B–D (**3**−**5**) were isolated from *O. bracteatum*. Their structures were characterized by spectroscopic analysis through 1D and 2D NMR, by MS spectroscopic analysis and comparing with those reported in the literatures. The anti-aging potential of the isolated benzoquinones was evaluated through a yeast lifespan assay, and the results indicated that **1**, **2**, **4** and **5** significantly extended the replicative lifespan of K6001 yeast, indicating that these benzoquinones obtained from *O. brateatum* have the ability to be employed as a potential therapeutic agent against age-related diseases.

## 1. Introduction

Aging is a natural process marked by progressive deterioration in physiological functions and increases in mortality and is often accompanied by various human pathologies, such as cardiovascular diseases, diabetes, stroke, cancer and neurodegenerative diseases (Parkinson’s and Alzheimer’s disease) [1]. Advances in public health practices, education and medicine have not only improved life expectancy but have also increased the aged population. According to the World Health Organization (WHO), the population of people aged 60 years will double by 2050 [2]. Increasing life expectancy also increases the prevalence of age-related diseases. Thus, strategies for discovering anti-aging molecules are of great importance [3].

Natural products have been an important source of drug discovery since ancient times, and a large number of present-day drugs are derived from natural sources. In our previous studies, we reported many natural compounds with anti-aging properties by using K6001 yeast replicative life span bioassay [4,5,6]. This assay is frequently used because yeast is inexpensive and has good reproducibility compared with other aging model organisms, such as vinegar flies, mice and nematodes [7,8,9,10]. In 2004, Jarolim et al., described a novel bioassay system with the K6001 yeast strain to improve the lifespan assay [11].

*Onosma bracteatum* Wall belongs to family Boraginaceae. It is known as Gaozaban in the Unani system of medicine and as Sedge in the Middle East. *O. bracteatum* is commonly used as a demulcent, alterative, diuretic, immunity enhancer and spasmolytic and as a major constituent of various Ayurvedic formulations for the treatment of hypertension, leprosy, rheumatism and asthma [12,13]. Pharmacological studies on *O. bracteatum* reported that it has antibacterial, analgesic, antioxidant and wound healing activities [14,15,16]. Additionally, it has acetyl-cholinesterase and NADH oxidase inhibitory activities. It also contains carbohydrates, fatty acids, flavonoids, tannins, glycosides and phenolic constituents [12,15].

The objective of this study was to obtain anti-aging compounds from *O. bracteatum* by using the K6001 yeast replicative life span bioassay. Anti-aging compounds (**1**, **2**, **4** and **5**) and inactive compound **3** were purified from *O. bracteatum* by column chromatography and then characterized by spectroscopic analysis and comparing with the data described in previously published reports (Figure 1).

## 2. Results

### 2.1. Isolation

The dried plant material of *O. bracteatum* was ground to uniform powder and extracted with methanol to obtain the crude extract. The crude extract was then partitioned between ethyl acetate and water. After bioassay evaluation of ethyl acetate and water layer samples, the active ethyl acetate layer sample was subjected to series of silica gel and ODS open column chromatography under the guidance of a bioassay system and was finally purified by HPLC to yield two known (**1** and **2**) and three novel (**3**–**5**) benzoquinone derivatives. The structures of known benzoquinones were identified as allomicrophyllone (**1**, 0.0043% of dry weight) [17,18] and ehretiquinone (**2**, 0.0013%) [19], while structures of novel benzoquinones were elucidated and named as ehretiquinone B (**3**, 0.00023%), ehretiquinone C (**4**, 0.00014%), and ehretiquinone D (**5**, 0.00028%) (Figure 1) by spectroscopic analysis, including 1D and 2D NMR, HR-ESI-TOF-MS and comparison of spectroscopic data with those reported in the literatures (see Appendix A).

### 2.2. Structure Elucidation

Ehretiquinone B (**3**) was obtained as a red powder and its molecular formula (C_22_H_20_O_5_) was determined by HR-ESI-TOF-MS. The ^1^H NMR data (Table 1) showed the presence of three aromatic protons at *δ*_H_ 6.53, 6.57 and 6.61, corresponding to a 1,2,4-trisubstituted benzene; two *cis* olefinic protons at *δ*_H_ 6.85 and 6.52; two *trans* olefinic protons at *δ*_H_ 6.18 and *δ*_H_ 5.68; three olefinic protons at *δ*_H_ 5.13, 5.31 and *δ*_H_ 5.64; two methylene protons at *δ*_H_ 2.52 and *δ*_H_ 2.78; one oxygenated methylene group at *δ*_H_ 4.20; one methine proton at *δ*_H_ 3.83 and one methyl group at *δ*_H_ 1.69. The ^13^C NMR data (Table 2) showed the presence of 22 carbon signals. These signals were attributed to two ketone groups (*δ*_C_ 195.4, 193.1), a 1,2,4-trisubstituted benzene (*δ*_C_ 114.3, 114.9, 117.6, 127.5, 144.9 and 150.0), eight olefinic carbons (*δ*_C_ 118.2, 122.5, 124.6, 131.9, 134.0, 138.7, 139.4 and 143.8), one oxygenated quaternary carbon (*δ*_C_ 80.6), one oxymethylene (*δ*_C_ 62.9), one quaternary carbon (*δ*_C_ 55.4), one methine (*δ*_C_ 39.3), one methylene (*δ*_C_ 36.2) and one methyl group (*δ*_C_ 22.7). The ^1^H-^1^H COSY spectra indicated the correlation signals between H-5 and H-6; H-7 and H-8; H-5′ and H-6′; H-7′ and H-8′. Based on these signals, a structural fragment of **3** was obtained, as shown in Figure 2. The HMBC spectrum indicated the following major ^1^H-^13^C correlation: H-3 to C-4; H-5 to C-1 and C-3; H-6 to C-1 and C-4; H-7 to C-1, C-3, C-9 and C-3′; H-8 to C-11; H-11 to C-8, C-9 and C-10; H-5′ to C-3′; H-6′ to C-2′ and C-4′; H-7′ to C-1′, C-2′, C-3′ and C-9′; H-8′ to C-9′ and H-11′ to C-9′ and C-10′. On the basis of these signals, the structure fragments were connected to obtain the planar structure of **3** as described in Figure 2.

The relative configuration of **3** was determined by NOESY correlations and coupling constants. The coupling constant (*J*_5′,6′_ = 10.5 Hz) indicated the *cis* configuration of protons at 5′ and 6′ position whereas the large coupling constant (*J*_7′,8′_ = 16.5 Hz) and NOESY correlation between H-11′/H-7′ (Figure 3) suggested the *trans* configuration of protons at 7′ and 8′ in **3**. In the NOESY spectrum, the correlation between H-11/H-8 confirmed the *trans* configuration of the double bond and H-10/H-7′ (Figure 3) indicated the same orientation of these protons and found it to be identical to ehretiquinone (**2**) [19]. In addition to same NOESY correlations, ^1^H NMR and ^13^C NMR of **3** and ehretiquinone (**2**), at 2′, 3′ and 7 positions, and close to these positions, were identical which also supported the identical relative configuration of **3** and ehretiquinone (**2**). While comparing **3** and ehretiquinone (**2**), the spectroscopic data suggested that the methyl group at C-11′ in ehretiquinone (**2**) was replaced by CH_2_OH in **3**. The remaining structure of **3** was found to be identical to ehretiquinone (**2**), as shown in Figure 1. Thus, the structure of **3** was determined and named as ehretiquinone B (Figure 1).

Ehretiquinone C (**4**) was obtained as a red powder. The molecular formula of **4** (C_22_H_20_O_5_) was determined by HR-ESI-TOF-MS. The ^1^H NMR and ^13^C NMR data of **4** (Table 1 and Table 2) showed that **4** has many similarities to ehretiquinone (**2**). The difference between the compounds was the replacement of CH_3_ (*δ*_H_ 1.70, *δ*_C_ 22.7) in **2** by CH_2_OH (*δ*_H_ 4.05, *δ*_C_ 65.8) in **4** at the 11 position, as shown in Figure 1. The difference was confirmed by the HMBC correlation between H-8 to C-11 and H-11 to C-8, C-9, and C-10, as shown in Figure 2. Meanwhile, the other ^1^H-^13^C correlations and the COSY correlation were identical to **3**. The NOESY correlation spectrum indicated that the relative configuration of **4** was similar to that of ehretiquinone B (**3**) (Figure 3). Thus, the structure of **4** was determined and named as ehretiquinone C (Figure 1).

Ehretiquinone D (**5**) was obtained as a yellow powder. The molecular formula of **5** (C_22_H_22_O_6_) was determined by HR-ESI-TOF-MS. The ^1^H NMR data and ^13^C NMR data (Table 1 and Table 2) were nearly identical to those of allomicrophyllone (**1**) [17,18]. The comparison of **5** and allomicrophyllone (**1**) indicated that CH_3_ (*δ*_H_ 1.16, *δ*_C_ 29.8) at the 10′ position in allomicrophyllone (**1**) was replaced by CH_2_OH (*δ*_H_ 3.41, *δ*_C_ 69.7) in **5** which was further confirmed by HMBC correlation among the positions H-8′ to C-10′; H-11′ to C-10′ and H-10′ to C-8′, as shown in Figure 2. The remaining ^1^H-^13^C HMBC and ^1^H-^1^H COSY correlations were found to be similar to those of ehretiquinone B (**3**) and ehretiquinone C (**4**) (Figure 2). The relative configuration of **5**, measured by the NOESY spectrum, was found to be similar to ehretiquinone B (**3**). However, the position 9′ of the side chain, attached at 2′, is the chiral center and methods for such stereochemistry are needed to be established, and the stereochemistry of position 9′ will remain to be clarified. Thus, the structure of **5** was determined and named as ehretiquinone D (Figure 1).

Allomicrophyllone (**1**) was obtained as a yellow powder. The molecular formula of **1** (C_22_H_22_O_5_) was determined by HR-ESI-TOF-MS and identified by comparing the MS, ^1^H NMR and ^13^C NMR data with the literature [17,18].

Ehretiquinone (**2**) was obtained as a red powder. The molecular formula of **2 (**C_22_H_20_O_4_**)** was determined by HR-ESI-TOF-MS and identified based on the comparison of MS, ^1^H NMR and ^13^C NMR data with those in the literature [19].

### 2.3. Anti-Aging Activity in K6001 Yeast Strain

Anti-aging activities of isolated benzoquinones (**1**–**5**) were evaluated by using a K6001 yeast replicative lifespan bioassay system. Resveratrol (Res), a well-known anti-aging substance, was used as a positive control to check the reliability of the bioassay system. The most effective concentration of Res was 10 μM in yeast replicative lifespan bioassay system as reported in our previous publications [4,5,6]. As shown in Figure 4, allomicrophyllone (**1**) at 3 μM, ehretiquinone (**2**) and ehretiquinone C (**4**) at 1 and 3 μM and ehretiquinone D (**5**) at 1 μM, exhibited the significant increase in the replicative lifespan of the K6001 yeast strain, while ehretiquinone B (**3**) was inactive. This result indicated the anti-aging potential of benzoquinones (**1**, **2**, **4** and **5**) isolated from *O. bracteatum.*

## 3. Discussion

In the context of global aging, research and development of anti-aging drugs to improve the quality of life of aged people and reduce the prevalence of age-related diseases, are of great importance. This study reports the isolation, structure elucidation, biological activity and brief SAR study of anti-aging benzoquinone derivatives from *O. bracteatum*.

In this study, two known (**1** and **2**) and three novel (**3**–**5**) benzoquinone derivatives were isolated from ethyl acetate layer sample of *O. bracteatum* under the guidance of K6001 yeast replicative lifespan bioassay. The structures of novel benzoquinones (**3**–**5**) were elucidated by extensive spectroscopic analysis. The known benzoquinones, allomicrophyllone (**1**) and ehretiquinone (**2**) were purified and identified by comparing spectroscopic data with those reported in the literatures [17,18,19]. Previously, both compounds were reported to have anti-allergic and anti-inflammatory activity, respectively [17,19]. To the best of our knowledge, no anti-aging related study was reported for these compounds. These isolated compounds shared the same benzoquinone skeleton and their differences were in the modifications at C-11 and C-11′ positions. 

The biological activities evaluation were performed for the study of structure activity relationship. Results in Figure 4 indicated that isolated allomicrophyllone (**1**), ehretiquinone (**2**), and ehretiquinones C–D (**4**–**5**) significantly prolonged the replicative lifespan of yeast. However, ehretiquinone B (**3**) which possessed the same benzoquinone skeleton but different modification was inactive. According to the structure activity relationship (SAR) study within these isolated compounds, we concluded that the presence of methyl and exomethylene groups at C-9′ is important for anti-aging activity as in ehretiquinone (**2**) and ehretiquinone C (**4**). The modification like the hydroxylation of C-11′ methyl group as in ehretiquinone B (**3**) resulted in the loss of activity. Furthermore, a comparison of ehretiquinone (**2**) and ehretiquinone C (**4**) with better activity than those of allomicrophyllone (**1**) and ehretiquinone D (**5**) showed that the presence of exomethylene group at C-9′ played an important role in anti-aging activity for these molecules. Ehretiquinone C (**4**) exhibited better anti-aging activity than that of resveratrol because the mean lifespan value of **4** at 1 µM is comparable with resveratrol at 10 µM which showed the same value. Therefore, ehretiquinone C (**4**) can be considered as a promising lead compound and its anti-aging mechanism of action should be further intensively studied. In addition, these results also provide important structure information to design and synthesize novel anti-aging drugs. This study will make a contribution to the prevention and treatment of age-related diseases. The results also suggested that *O. bracteatum* enriched with anti-aging ingredients can be used as medicine or as a food supplement.

## 4. Materials and Methods

### 4.1. General

A JASCO P-1030 digital polarimeter was used for optical rotation measurements. Preparative high-performance liquid chromatography (HPLC) with a system equipped with ELITE P-230 pumps and a UV detector was used. High-resolution mass spectra (HR-ESI-TOF-MS) were obtained with an Agilent Technologies 6224A accurate mass TOF LC/MS system. NMR spectra were observed with Bruker AV III-500 spectrometer and NMR chemical shifts in *δ* (ppm) were referenced to solvent peaks of *δ*_C_ 77.0 and *δ*_H_ 7.26 for CDCl_3_. Column chromatography was performed by using silica gel (200–300 mesh, Yantai Chemical Industry Research Institute, Yantai, China) or reversed phase C18 (Octadecylsilyl, ODS) silica gel (Cosmosil 75C18-OPN, Nacalai Tesque, Kyoto, Japan).

### 4.2. Plant Material

The plant material was purchased from Mansehra, Khyber Pakhtunkhwa, Pakistan and identified as *Onosma bracteatum* Wall by Dr. Zafar Ullah Zafar, Associate Professor, Institute of Pure and Applied Biology, Bahauddin Zakariya University, Multan, Pakistan. The voucher specimen (20170220) of the plant was deposited at the Institute of Materia Medica, Zhejiang University.

### 4.3. Extraction and Isolation

Dried plant material (1.5 kg) was ground to powder and extracted with methanol (CH_3_OH) for 3 days along with continuous shaking at room temperature. The obtained crude extract (120 g) was partitioned between ethyl acetate (EtOAc) and water (H_2_O). The active ethyl acetate layer sample (30 g) was chromatographed on a silica gel open column and eluted with *n*-hexane/CH_2_Cl_2_ (100:0, 80:20, 50:50, 0:100) and CH_2_Cl_2_/CH_3_OH (98:2, 95:5, 90:10 and 0:100). Six fractions were obtained. Fraction 4 (8 g) that was eluted with CH_2_Cl_2_/CH_3_OH (95:5, 90:10) was further processed by ODS open column. CH_3_OH/H_2_O (40:60, 50:50, 55:45, 60:40, 70:30, 90:10 and 100:0) was used as solvent. Of the seven fractions (Fr.1–Fr.7) obtained, Fr.2–Fr.4, were further separated as follows:

Fr.4 (500 mg) was separated by silica gel open column and eluted with CH_2_Cl_2_/CH_3_OH (100:0, 99:1, 98:2, 97:3, 95:5, 90:10 and 0:100). Nine samples (Fr.4-1–Fr.4-9) were obtained. Sample Fr.4-3 was further purified by HPLC [C30-UG-5 (ϕ10 × 250 mm, Nomura Chemical), mobile phase: acetonitrile (CH_3_CN)/H_2_O (40:60), flow rate: 3 mL/min, and detector: 210 nm] and yielded compound (**1**) (65 mg, *t*_R_ = 21.2 min).

Fr.3 (113 mg) was separated by using ODS open column and eluted with CH_3_OH/H_2_O (40:60, 45:55, 50:50, 55:45, 60:40, 80:20 and 100:0). Ten fractions (Fr.3-1–Fr.3-10) were obtained, and Fr.3-7 was further purified by HPLC [C30-UG-5 (ϕ10 × 250 mm, Nomura Chemical), mobile phase: methanol (CH_3_OH)/H_2_O (68:32), flow rate: 3 mL/min, and detector: 210 nm] and yielded compound (**2**) (20 mg, *t*_R_ = 20.3 min).

Fr.2 (450 mg) was subjected to silica gel open column and eluted with CH_2_Cl_2_:CH_3_OH (100:0, 99:1, 98:2, 97:3, 95:5, 90:10, 0:100). Nine fractions (Fr.2-1–Fr.2-9) were obtained, and Fr.2-4 (20 mg) was subjected to ODS open column chromatography with CH_3_OH/H_2_O (30:70, 32:68, 35:65, 40:60 and 100:0), and the fifth fraction (6.5 mg) was purified by HPLC [C30-UG-5 (ϕ10 × 250 mm, Nomura Chemical), mobile phase: acetonitrile (CH_3_CN)/H_2_O (35:65), flow rate: 3 mL/min, and detector: 210 nm] to yield compound (**3**) (3.5 mg, *t*_R_ = 31.0). Fr.2-5 (70 mg) was separated by ODS open column using CH_3_OH/H_2_O (25:75, 27:73, 30:70, 32:68, 34:66, 36:64, 40:60 and 100:0) and the seventh fraction (11 mg) was purified by HPLC [C30-UG-5 (ϕ10 × 250 mm, Nomura Chemical), mobile phase: acetonitrile (CH_3_CN)/H_2_O (32:68), flow rate: 3 mL/min, and detector: 210 nm] to yield compound (**4**) (2.1 mg, *t*_R_ = 43.3). Fr.2-8 (75 mg) was separated by ODS open column using CH_3_OH/H_2_O (20:80, 22:78, 24:76, 25:75, 26:74, 30:70, 35:65, 40:60 and 100:0) and the seventh fraction (10 mg) was purified by HPLC [C30-UG-5 (ϕ10 × 250 mm, Nomura Chemical), mobile phase: acetonitrile (CH_3_CN)/H_2_O (30:70), flow rate: 3 mL/min, and detector: 210 nm] to yield compound (**5**) (4.3 mg, *t*_R_ = 22.3).

*Allomicrophyllone* (**1**). Yellow powder; ${[\alpha ]\;}_{D}^{16}$ + 0.02 (c 0.57, CHCl_3_); High-resolution ESI-TOF-MS *m*/*z* 389.1359, calculated for C_22_H_22_O_5_Na (M + Na)^+^ 389.1359; ^1^H NMR (500 MHz, CDCl_3_): *δ*_H_ = 6.83 (1H, d, *J* = 10.5 Hz, H-5′), 6.61 (1H, d, *J* = 8.6 Hz, H-6), 6.56 (1H, d, *J* = 2.8 Hz, H-3), 6.51 (1H, dd, *J* = 2.8, 8.6 Hz, H-5), 6.51 (1H, d, *J* = 10.5 Hz, H-6′), 5.76 (1H, d, *J* = 15.9 Hz, H-8′), 5.67 (1H, d, *J* = 15.9 Hz, H-7′), 5.65 (1H, d, *J* = 7.1 Hz, H-8), 3.77 (1H, d, *J* = 7.1 Hz, H-7), 2.73 (1H, d, *J* = 19.2 Hz, Hb-10), 2.50 (1H, d, *J* = 19.2 Hz, Ha-10), 1.69 (3H, s, H-11), 1.22 (6H, s, H-10′ & H-11′); ^13^C NMR (125 MHz, CDCl_3_): *δ*_C_ = 195.8 (C-1′), 193.1 (C-4′), 150.0 (C-4), 145.0 (C-1), 143.3 (C-8′), 139.4 (C-5′), 138.6 (C-6′), 131.8 (C-9), 127.7 (C-2), 122.5 (C-8), 122.0 (C-7′), 117.6 (C-6), 114.9 (C-5), 114.4 (C-3), 80.7 (C-3′), 71.0 (C-9′), 54.9 (C-2′), 39.1 (C-7), 36.2 (C-10), 29.8 (C-10′), 29.8 (C-11′), 22.6 (C-11). The structure was identified through comparison of MS, ^1^H NMR and ^13^C NMR spectra as well as specific rotation with literatures [17,18].

*Ehretiquinone* (**2**). Red powder; ${[\alpha ]\;}_{D}^{16}$ + 1.01 (c 0.12, CH_3_OH); high-resolution ESI-TOF-MS *m*/*z* 371.1254, calculated for C_22_H_20_O_4_Na (M + Na)^+^ 371.1252; ^1^H NMR (500MHz, CDCl_3_): *δ*_H_ = 6.83 (1H, d, *J* = 10.5 Hz, H-5′), 6.62 (1H, d, *J* = 8.6 Hz, H-6), 6.58 (1H, d, *J* = 2.2 Hz, H-3), 6.53 (1H, m, H-5), 6.52 (1H, d, *J* = 10.5 Hz, H-6′), 6.25 (1H, d, *J* = 16.0 Hz, H-8′), 5.65 (1H, d, *J* = 6.2 Hz, H-8), 5.50 (1H, d, *J* = 16.0 Hz, H-7′), 5.04 (1H, s, Hb-10′), 4.97 (1H, s, Ha-10′), 3.83 (1H, d, *J* = 6.2 Hz, H-7), 2.78 (1H, d, *J* = 19.1 Hz, Hb-10), 2.52 (1H, d, *J* = 19.1 Hz, Ha-10), 1.73 (3H, s, H-11′), 1.70 (3H, s, H-11); ^13^C NMR (125 MHz, CDCl_3_): *δ*_C_ = 195.4 (C-1′), 193.2 (C-4′), 150.0 (C-4), 145.1 (C-1), 140.8 (C-9′), 139.2 (C-5′), 138.8 (C-6′), 137.5 (C-8′), 131.9 (C-9), 127.7 (C-2), 124.1 (C-7′), 122.7 (C-8), 119.3 (C-10′), 117.6 (C-6), 114.9 (C-5), 114.4 (C-3), 80.7 (C-3′), 55.3 (C-2′), 39.4 (C-7), 36.3 (C-10), 22.7 (C-11), 18.3 (C-11′). The structure was identified through comparison of MS, ^1^H NMR and ^13^C NMR spectra as well as specific rotation with literature [19].

*Ehretiquinone B* (**3**). Red powder; ${[\alpha ]\;}_{D}^{16}$ + 0.99 (c 0.31, CH_3_OH); high-resolution ESI-TOF-MS *m*/*z* 387.1208, calculated for C_22_H_20_O_5_Na (M + Na)^+^ 387.1203; Data for ^1^H NMR and ^13^C NMR are described in Table 1 and Table 2 respectively.

*Ehretiquinone C* (**4**). Red powder; ${[\alpha ]\;}_{D}^{16}$ + 0.37 (c 0.12, CH_3_OH); high-resolution ESI-TOF-MS *m*/*z* 387.1174, calculated for C_22_H_20_O_5_Na (M + Na)^+^ 387.1174; Data for ^1^H NMR and ^13^C NMR are described in Table 1 and Table 2 respectively.

*Ehretiquinone D* (**5**). Yellow powder; ${[\alpha ]\;}_{D}^{16}$ + 0.51 (c 0.41, CHCl_3_); high-resolution ESI-TOF-MS *m*/*z* 405.1300, calculated for C_22_H_22_O_6_Na (M + Na)^+^ 405.1300; Data for ^1^H NMR and ^13^C NMR are described in Table 1 and Table 2 respectively.

### 4.4. Lifespan Assay

The bioassay method used for the anti-aging study was previously described [4]. K6001 yeast strain was cultured on a YPGalactose or YPGlucose medium. YPGalactose medium was prepared with 3% galactose, 2% hipolypeptone and 1% yeast extract while YPGlucose medium, which contained 2% glucose instead of galactose. To prepare agar plates, 2% agar was added to the medium. To evaluate samples through bioassay, the K6001 yeast strain was inoculated and incubated for 24 h in the yeast peptone galactose medium along with shaking. After 24 h, the medium containing the yeast strain was centrifuged. The resulting yeast pellets were washed with phosphate buffer solution (PBS) three times and diluted. After dilution, the cells were calculated with a hemocytometer and around 4000 cells inoculated on agar plates containing samples with various concentrations. Then, the plates were placed in an incubator at 28 °C for 2 days. The yeast cells in the agar plate were observed under a microscope after 48 h. A total of 40 microcolonies from each plate were selected randomly, and the number of daughter cells were counted and analyzed.

### 4.5. Statistical Analysis

The significant differences among groups were determined by one-way ANOVA with Dunnett’s multiple comparison test. GraphPad Prism 5.0 software was used. A *p* value of < 0.05 was considered statistically significant.

## 5. Conclusions

Three new and two known benzoquinones were isolated and characterized from *O. bracteatum* under the guidance of K6001 yeast replicative lifespan assay. The structure of the new benzoquinones (**3**–**5**) were elucidated by spectroscopic analysis, and the known ones (**1**–**2**) were identified by comparing the spectroscopic data with those in literatures. All the isolated benzoquinones were evaluated for anti-aging activity by yeast replicative lifespan assay and **1**, **2**, **4** and **5** exhibited significant anti-aging activity. Ehretiquinone C (**4**) showed the most promising anti-aging activity and can thus be considered a potential lead compound. Its anti-aging mechanism should be further studied.

## Figures and Tables

**Figure 1 molecules-24-01428-f001:**
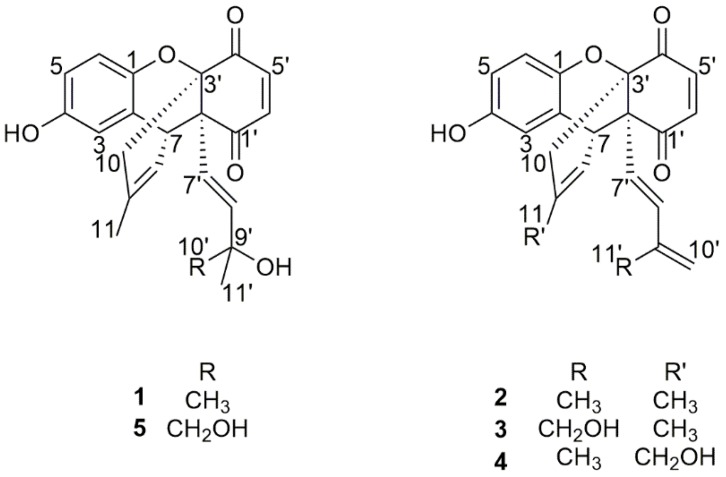
Chemical structures of allomicrophyllone (**1**), ehretiquinone (**2**) and ehretiquinones B–D (**3**–**5**).

**Figure 2 molecules-24-01428-f002:**
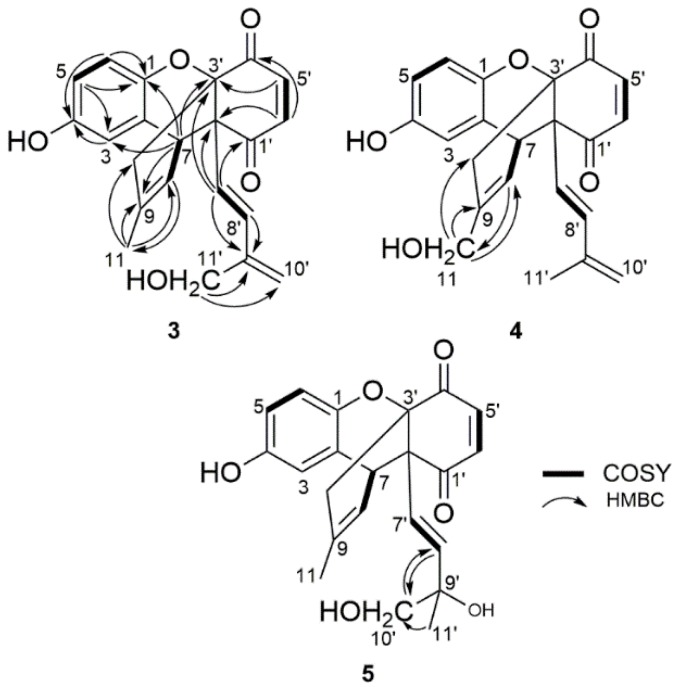
Gross structure of ehretiquinones B–D (**3**–**5**) with ^1^H-^1^H COSY and selected HMBC correlations.

**Figure 3 molecules-24-01428-f003:**
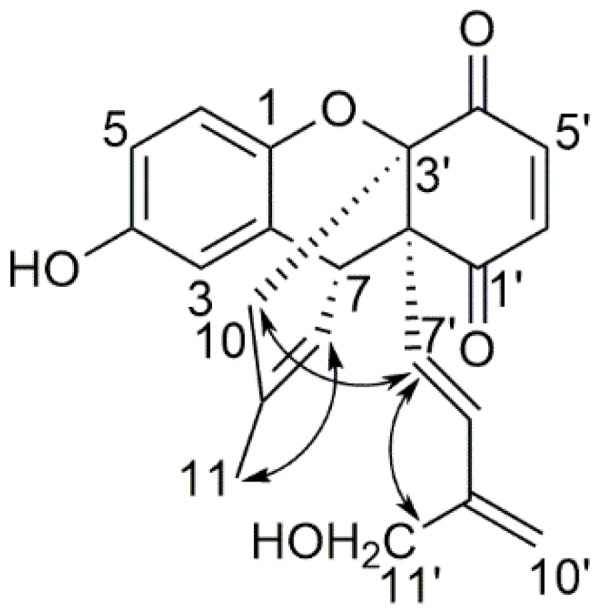
Selected NOESY correlations of ehretiquinone B (**3**).

**Figure 4 molecules-24-01428-f004:**
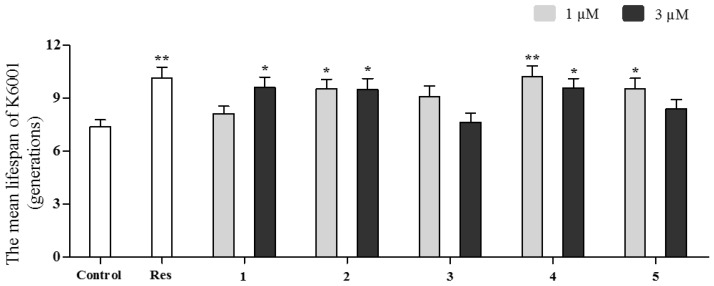
Effect of allomicrophyllone (**1**), ehretiquinone (**2**) and ehretiquinones B–D (**3**–**5**) on the replicative lifespan of the K6001 yeast strain. The average lifespan of K6001 was as follows: control (7.37 ± 0.41); Res at 10 μM (10.15 ± 0.61 **); **1** at 1 μM (8.12 ± 0.41) and at 3 μM (9.62 ± 0.57 *); **2** at 1 μM (9.52 ± 0.55 *) and at 3 μM (9.50 ± 0.61 *); **3** at 1 μM (9.10 ± 0.59) and at 3 μM (7.65 ± 0.50); **4** at 1 μM (10.22 ± 0.58 **) and at 3 μM (9.57 ± 0.54 *); **5** at 1 μM (9.52 ± 0.60 *) and at 3 μM (8.40 ± 0.53) (* *p* < 0.05 and ** *p* < 0.01, compared with the control).

**Table 1 molecules-24-01428-t001:** ^1^H NMR (500 MHz) data of ehretiquinones B–D (**3**–**5**) (CDCl_3_, *δ*_H_ in ppm, *J* in Hz).

Proton	3	4	5
3	6.57 (1H, d, *J* =2.9)	6.60 (1H, d, *J* = 2.8)	6.59 (1H, d, *J* = 2.70)
5	6.53 (1H, m)	6.53 (1H, m)	6.55 (1H, m)
6	6.61 (1H, d, *J* = 8.6)	6.62 (1H, d, *J* = 8.7)	6.63 (1H, d, *J* = 8.6)
7	3.83 (1H, d, *J* = 6.3)	3.95 (1H, d, *J* = 6.3)	3.78 (1H, d, *J* = 6.2)
8	5.64 (1H, d, *J* = 6.3)	5.95 (1H, d, *J* = 5.9)	5.6 (1H, m)
10	2.52 (1H, d, *J* = 19.2) 2.78 (1H, d, *J* = 19.2)	2.63 (1H, d, *J* = 19.2) 2.84 (1H, d, *J* = 19.1)	2.51 (1H, d, *J* = 19.2) 2.73 (1H, d, *J* = 19.1)
11	1.69 (3H, s)	4.05 (2H, s)	1.69 (3H, s)
5′	6.85 (1H, d, *J* = 10.5)	6.85 (1H, d, *J* = 10.5)	6.89 (1H, d, *J* = 10.4)
6′	6.52 (1H, m)	6.52 (1H, d, *J* = 10.5)	6.52 (1H, d, *J* = 10.4)
7′	5.68 (1H, d, *J* = 16.5)	5.49 (1H, d, *J* = 16.1)	5.80 (1H, d, *J* = 15.9)
8′	6.18 (1H, d, *J* = 16.5)	6.26 (1H, d, *J* = 16.0)	5.62 (1H, d, *J* = 15.9)
10′	5.13 (1H, s) 5.31 (1H, s)	4.98 (1H, s) 5.05 (1H, s)	3.41 (2H, m)
11′	4.20 (2H, s)	1.72 (3H, s)	1.16 (3H, s)

**Table 2 molecules-24-01428-t002:** ^13^C NMR (125 MHz) data of ehretiquinones B–D (**3**–**5**) (CDCl_3_, *δ*_C_ in ppm).

No	3	4	5
1	144.9	145.0	144.9
2	127.5	126.9	127.8
3	114.3	114.5	114.2
4	150.0	150.1	150.2
5	114.9	115.1	115.1
6	117.6	117.7	117.8
7	39.3	38.9	39.2
8	122.5	122.9	122.3
9	131.9	135.1	132.0
10	36.2	32.2	36.1
11	22.7	65.8	22.7
1′	195.4	195.1	195.0
2′	55.4	55.5	55.2
3′	80.6	80.5	80.6
4′	193.1	192.9	192.9
5′	139.4	139.2	140.0
6′	138.7	138.8	138.5
7′	124.6	123.5	125.3
8′	134.0	137.7	139.4
9′	143.8	140.7	73.5
10′	118.2	119.7	69.7
11′	62.9	18.3	24.0

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
