# Peer review of "Novel Anti-Aging Benzoquinone Derivatives from Onosma bracteatum Wall"

_molecules, 2019, doi:10.3390/molecules24071428_

Round 1
Reviewer 1 Report
Dear authors
you paper entitled Novel Anti-Aging Benzoquinone Derivatives from Onosma bracteatum Wall is quite interesting to the scientific community but also to broad spectra since dealing with an interesting subject - ageing. However some conclusions you made are not supported by presented results or need to be explained.
In the 2.3. section you describe the results obtained on the replicate lifespan of the yeast and indicate that ALL used substances (detected in the plant) show strong anti-aging potential using t-test.
- How do you explain that some larger concentrations are not "working" while lower do (for some substances) while at 1 are opposite (and expected)?
- I strongly recommend to use ANOVA with Dunnetts multiple comparison test.
- You do not mention what Res is (although you mention it in Figure legend it is not clear what it is (and I have not found it in material section as well or I just missed it).
- Please clearly indicate in the Figure legend what numbers (1-5) mean since Figure should stand alone and if one do not have text it can not be followed.
- Complete section "we inferred that ehretiquinone (2) and ehretiquinone C (4) showed 158 increased activity because of the presence of one methyl group at C-11’ and one methylene at C10’. 159 Meanwhile, in ehretiquinone B (3), methyl groups at C11’ were hydroxylated and activity is reduced 160 as compared to those in ehretiquinone (2). Similarly, a comparison of ehretiquinone (2) with 161 allomicrophyllone (1) and ehretiquinone D (5) showed that methylene group at C-10’ was reduced 162 or hydroxylated and C-9 was hydroxylated, and thus the anti-aging activity was weak. Thus, 163 ehretiquinone (2) and ehretiquinone C (4) can be considered as a promising" needs to be rewritten! How do you explain the notification that 1 is working less than 5 when this can not be seen on Figure nor do you have statistics for it. Use Tukey test to try to prove your claims (e.g. 2 is working equally as 1 and 5).
Reviewer 2 Report
Discussion and hypothesis of manuscript are very weak. Current presentation of data is still incomplete for a scientific article
Reviewer 3 Report
Please find my comments about the results and the presentation below:
· The first sentence of the abstract should be removed.
· What do the dashed lines mean in Figure 1.
· The "Isolation" chapter is unacceptable. You need to expand it. In addition, the sentence can not start with a chemical formula.
· The first paragraph of the "Structure elucidation" chapter is very difficult to analyze. The data in brackets should be deleted eg (1H, d, J = 8.6 Hz)- it is unnecessary here.
· Table 1: compound 5: why the continuous couplings of the protons at 5 'and 6' carbons do not agree.
· What does the term "active ethyl acetate" mean?
· Please standardize the recording of organic solvents in the text. e.g. MeOH or CH3OH
· The spectroscopic data obtained for compounds 1 and 2 should be included.
· In the supplementary data, the spectra of new compounds should be included.
Round 2
Reviewer 2 Report
The manuscript was revised and re-written taking into consideration the reviewer’s suggestions and comments. The current version has been improved. Therefore, It is suitable for publication.
Reviewer 3 Report
I don't have no more questions for authors.